Applied and Environmental Science

# Impact of Glyphosate on the Honey Bee Gut Microbiota: Effects of Intensity, Duration, and Timing of Exposure

Erick V. S. Motta,[a] Nancy A. Moran[a]

aDepartment of Integrative Biology, University of Texas at Austin, Austin, Texas, USA

**ABSTRACT** Exposure to anthropogenic chemicals may indirectly compromise animal health by perturbing the gut microbiota. For example, the widely used herbicide glyphosate can affect the microbiota of honey bees, reducing the abundance of beneficial bacterial species that contribute to immune regulation and pathogen resistance. Previous studies have not addressed how this impact depends on concentration, duration of exposure, or stage of microbiota establishment. Worker bees acquire their microbiota from nestmates early in adult life, when they can also be exposed to chemicals collected by foragers or added to the hives. Here, we investigated how the gut microbiota of honey bees is affected by different concentrations of glyphosate and compared the effects with those caused by tylosin, an antibiotic commonly used to treat hives. We treated newly emerged workers at the stage at which they acquire the microbiota and also workers with established gut microbiota. Treatments consisted of exposure to sucrose syrup containing glyphosate in concentrations ranging from 0.01 mM to 1.0 mM or tylosin at 0.1 mM. Based on 16S rRNA amplicon sequencing and quantitative PCR (qPCR) determination of abundances, glyphosate perturbed the gut microbiota of honey bees regardless of age or period of exposure. *Snodgrassella alvi* was the most affected bacterial species and responded to glyphosate in a dose-dependent way. Tylosin also perturbed the microbiota, especially at the stage of acquisition, and the effects differed sharply from the effects of glyphosate. These findings show that sublethal doses of glyphosate (0.04 to 1.0 mM) and tylosin (0.1 mM) affect the microbiota of honey bees.

**IMPORTANCE** As is true of many animal species, honey bees depend on their gut microbiota for health. The bee gut microbiota has been shown to regulate the host immune system and to protect against pathogenic diseases, and disruption of the normal microbiota leads to increased mortality. Understanding these effects can give broad insights into vulnerabilities of gut communities, and, in the case of honey bees, could provide information useful for promoting the health of these economically critical insects, which provide us with crop pollination services as well as honey and other products. The bee gut microbiota is acquired early in adult life and can be compromised by antibiotics and other chemicals. The globally used weed killer glyphosate was previously found to impact the gut microbiota of honey bees following sustained exposure. In the present study, we address how this impact depends on concentration, duration of exposure, and stage of community establishment. We found that sublethal doses of glyphosate reduce the abundance of beneficial bacteria and affect microbial diversity in the guts of honey bees, regardless of whether exposure occurs during or after microbiota acquisition. We also compared the effects of glyphosate to those of tylosin, an antibiotic used in beekeeping, and observed that tylosin effects diverge from those caused by glyphosate and are greater during microbiota acquisition. Such perturbations are not immediately lethal to bees but, depending on exposure level, can decrease survivorship under laboratory conditions.

Address correspondence to Erick V. S. Motta, erickvsm@utexas.edu.

More evidence that the herbicide glyphosate perturbs the beneficial honey bee gut microbiota. @ErickMotta19 and @NancyMoran15 show this impact is dose dependent, and occurs regardless of duration of exposure or stage of community establishment.

**KEYWORDS** *Apis mellifera*, antibiotic, herbicide, microbiome, *Snodgrassella alvi*

The gut microbiota is increasingly recognized as a key factor in the health of animal hosts (1). This is the case for the western honey bee, *Apis mellifera*, which harbors a specialized gut microbiota that is required for normal development and nutrition and for protection against pathogens (2–5). Honey bees are the most important agricultural pollinator, but they have been undergoing alarming increases in colony failure during the last decade. Several factors implicated in these declines include parasites, pathogens, poor nutrition, and pesticides (6, 7). Recent studies have demonstrated that some agrochemicals can perturb the honey bee gut microbiota and thereby compromise bee health. For example, honey bee hives can be directly exposed to antibiotics, such as oxytetracycline and tylosin, which are used to treat or prevent colony infections by bacterial pathogens (8). This exposure can reduce the abundance of beneficial bacteria in the adult bee gut, thus increasing susceptibility to infection by opportunistic pathogens (9, 10).

More recently, glyphosate, the primary herbicide used globally for weed control, has been linked to perturbation of the gut microbiota of honey bees (11–13). Glyphosate inhibits an enzyme, 5-enolpyruvyl-shikimate-3-phosphate synthase (EPSPS), in the shikimate pathway, found not only in plants but also in most microorganisms, including those in the bee gut, such as *Snodgrassella alvi*, *Gilliamella* spp., and *Bifidobacterium* spp. (13). Organisms possessing the shikimate pathway rely on it for the production of essential aromatic compounds, such as the amino acids tryptophan, phenylalanine, and tyrosine, which may be unavailable in the environment. In honey bees, protein digestion and amino acid absorption occur primarily in the midgut (14). Thus, amino acids derived from the diet may not reach the hindgut compartments (ileum and/or rectum), which are the regions containing more than 95% of the bee gut microbiota (15, 16). Compared to bees lacking a gut microbiota, bees with an established gut microbiota exhibit higher concentrations of the aromatic amino acids in the hindgut (3, 17). A mutagenic screen of *Snodgrassella* revealed that amino acid biosynthetic pathways are required to colonize hosts, indicating that amino acids are limiting in the hindgut, having been efficiently absorbed by the host midgut epithelium (18). Colonization by *Snodgrassella* and other species able to synthesize all amino acids may foster the establishment of *Lactobacillus* species from Firm-4 and Firm-5 clades, which cannot synthesize some essential amino acids (19). Thus, inhibition of the shikimate pathway may deplete essential nutrients required by the hindgut community as a whole, culminating in bacterial death.

Indeed, in honey bees, glyphosate exposure has been shown to reduce abundance of some beneficial bacteria, such as *Snodgrassella*, a Gram-negative bacterium that forms a biofilm on the ileum wall (13). Therefore, glyphosate can act as an environmental stressor that indirectly affects honey bees by perturbing the gut microbiota. However, understanding whether and when colonies undergo such perturbation depends on knowledge of its dependence on dose, timing, and duration of exposure. Honey bees acquire their microbiotas in the first few days of adult life, after which the composition remains largely stable (2). The effects of antimicrobial agents, such as glyphosate, may vary based on the stage of microbial community establishment. While foraging bees are more likely to suffer brief topical and/or oral exposure to higher concentrations of glyphosate, young worker bees, at the stage of microbial acquisition or with established microbial communities, tend to be chronically exposed to lower doses of the herbicide present in hive compartments. A semi-field experiment quantified glyphosate in nectar and pollen collected by foraging bees from plants recently treated with a glyphosate-based formulation, with concentrations ranging from residual to 31.3 mg/kg and 629 mg/kg, respectively (20). In the same experiment, glyphosate concentrations in nectar samples taken from colonies ranged from residual to 1.30 mg/kg. Glyphosate has also been detected in commercial (up to 163 $\mu$g/kg) and raw (up to 342 $\mu$g/liter) honey (21, 22) and in other environmental matrices, such as water (up to

mSystems®

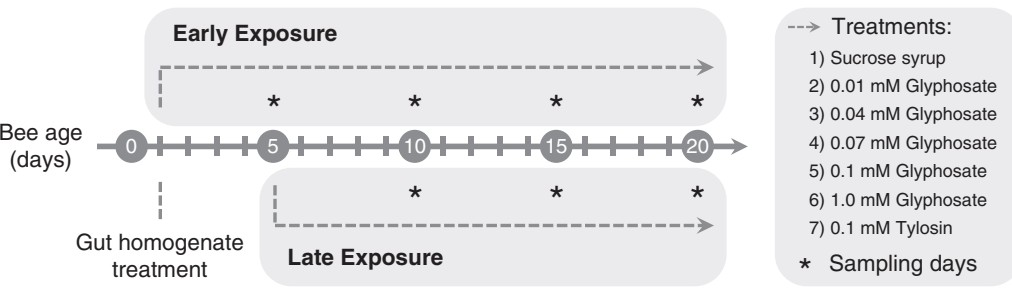

**FIG 1** Chronic exposure of recently emerged honey bees to glyphosate or tylosin. Recently emerged bees were transferred to cup cages containing sterile sucrose syrup and bee bread mixed with bee gut homogenate. Then, they were divided into two main groups, 1-day-old bees (early exposure) and 5-day-old bees (late exposure), each split into 7 subgroups fed sterile sucrose syrup with no additive (controls) or with 0.01, 0.04, 0.07, 0.1, or 1.0 mM glyphosate or 0.1 mM tylosin for 20 days and 15 days, respectively. Bees were sampled from each group every 5 days for the period of exposure.

3.1 mg/liter) (23–25), sediments (up to 6.8 mg/kg) (26, 27), soil (up to 5.0 mg/kg) (27), and foliage (up to 448 mg/kg) (28). These point-located studies may not represent the overall use of glyphosate in crop and noncrop areas; thus, little is known about how often and how much bees are exposed to the herbicide.

Here, we investigate the effects of glyphosate on the honey bee gut microbiome by exposing newly emerged workers, both during and after microbiota acquisition, and using different concentrations of glyphosate, ranging from residues detected in the environment (23–25) to those detected in nectar and pollen in a semi-field experiment (20). We also investigated and compared the effects of tylosin, an antibiotic commonly used in beekeeping, to the effects of glyphosate. For that, we used quantitative PCR (qPCR) and 16S rRNA gene sequence analyses to measure bacterial abundance and composition in the bee guts. Our findings show for the first time that glyphosate-mediated perturbation of the bee gut microbiota, especially the effects on *Snodgrassella* abundance, is dose dependent and that a similar pattern of perturbation occurs regardless of the age of the bee and duration of exposure.

## RESULTS

**Effects of glyphosate and tylosin on honey bees during and after acquisition of the gut microbiota.** Two groups of recently emerged bees, 1-day-old bees still acquiring the gut microbiota (early-exposure group) and 5-day-old bees with established gut microbiotas (late-exposure group), were each divided into seven subgroups to be continually exposed to five different concentrations of glyphosate ranging from 0.01 mM (1.691 mg/liter, or 1.375 mg/kg) to 1.0 mM (169.1 mg/liter, or 137.5 mg/kg), tylosin at 0.1 mM (91.61 mg/liter, or 74.48 mg/kg), or sucrose syrup only. Because we expected tylosin, as an antibacterial compound, to affect the community, this treatment served as a type of positive control. Bees were sampled at different times after the start of treatment (days 5, 10, 15, and 20 in the early-exposure group and days 5, 10 and 15 in the late-exposure group) (Fig. 1). RNA was extracted from whole guts of sampled bees and processed for 16S rRNA amplicon sequencing.

Exploratory principal-component analysis (PCA) of the relative abundance of the main bee gut bacterial taxa suggested a continuum in divergence between gut communities of control and glyphosate-treated bees in both early- and late-exposure groups, with higher glyphosate concentrations and later sampling dates being associated with greater divergence (Fig. 2). The taxa contributing most to this divergence were *Snodgrassella* and *Gilliamella*, which were negatively correlated with glyphosate-treated bees in both exposure groups, and *Bifidobacterium*, *Lactobacillus* Firm-4, and *Lactobacillus* Firm-5, which were positively correlated with glyphosate-treated bees, especially in the early-exposure group. Noncore environmental bacteria were also positively associated with glyphosate-treated samples in the early-exposure group.

Communities in tylosin-treated bees diverged from those in both control and glyphosate-treated bees throughout the experiment in the early-exposure group and

mSystems®

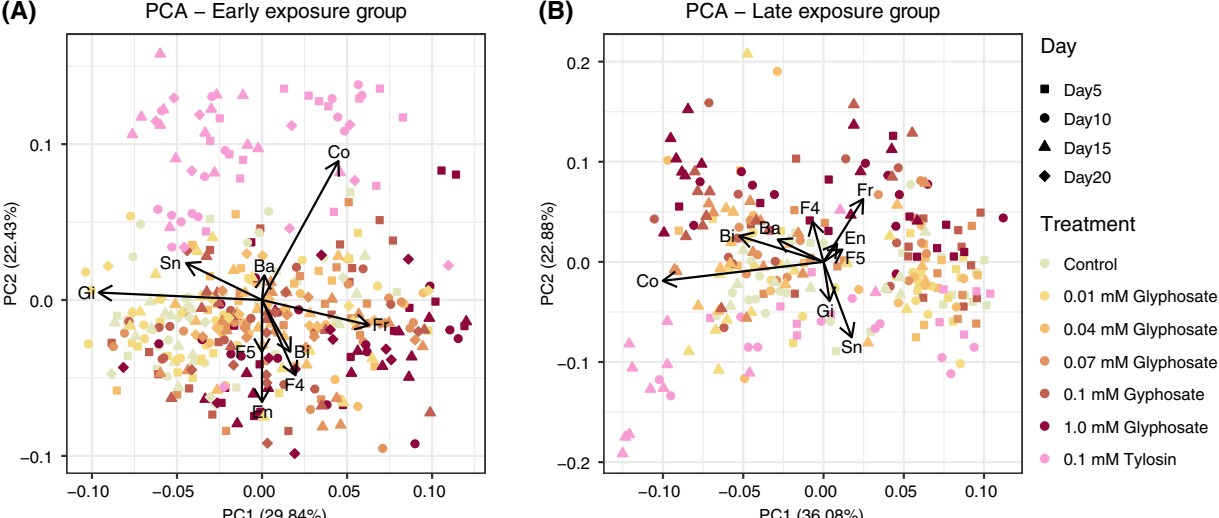

**FIG 2** Projection of the relative abundances of bacterial taxa into the first and second principal components in sampled bees, together with correlation vectors representing variables driving the separation on both axes. (A) Bees sampled from the early-exposure group at days 5, 10, 15, and 20 after the start of treatment. (B) Bees sampled from the late-exposure group at days 5, 10, and 15 after the start of treatment. Ba, *Bartonella*; Bi, *Bifidobacterium*; Co, *Commensalibacter*; En, environmental bacteria; Fr, *Frischella*; Gi, *Gilliamella*; F4, *Lactobacillus* Firm-4; F5, *Lactobacillus* Firm-5; Sn, *Snodgrassella*.

to a lesser extent in the late-exposure group (Fig. 2). They showed shifts opposite to those in glyphosate-treated bees in relative abundances of bacterial taxa. Moreover, *Commensalibacter* abundance was positively correlated with tylosin-treated bees at specific sampling times in both early- and late-exposure groups. Some of these trends from the PCAs correspond to significant changes in bacterial relative abundance between control and treated samples (Fig. S1; Table S1A).

We also estimated absolute bacterial abundance using quantitative PCR (qPCR). Glyphosate treatment reduced total 16S rRNA copy numbers in the guts of bees exposed to 1.0 mM glyphosate at day 20 in the early-exposure group, while absolute abundances were not significantly affected for lower concentrations, earlier sampling times, or late exposure (Fig. 3; Table S1B). Taking into consideration both relative and absolute abundances, glyphosate treatment inhibited growth of *Snodgrassella*, for which 16S rRNA copy numbers were reduced in all bees treated with 0.07 to 1.0 mM glyphosate throughout the experiment in both early- and late-exposure groups (Fig. 4; Table S1C). In most cases, the 0.04 mM treatment also resulted in a significant decrease in *Snodgrassella*. A dose-dependent response of *Snodgrassella* abundance to glyphosate treatment was observed for both early- and late-exposure groups (Fig. 5). The 50% effective doses (ED$_{50}$s), i.e., the half-maximal effective concentrations of glyphosate at which *Snodgrassella* growth is reduced by 50%, ranged from 0.02 mM to 0.06 mM glyphosate (Table S2). Regarding other core bee gut bacteria, *Gilliamella* absolute numbers were reduced in bees treated with 0.07 mM or 1.0 mM glyphosate at specific sampling times in the early-exposure group (Fig. S2; Table S1C).

Tylosin treatment broadly impacted the gut microbiome in the early-exposure group, decreasing total bacterial abundance (Fig. 3A; Table S1B) and specifically lowering abundance of the Gram-negative bacteria *Snodgrassella* (days 5, 10, and 20 [Fig. 4A; Table S1C]) and *Gilliamella* (day 5 [Fig. S2A; Table S1C]), as well as the Gram-positive bacteria *Lactobacillus* Firm-5, *Lactobacillus* Firm-4, and *Bifidobacterium* at almost all sampling times (Fig. S2; Table S1C). However, in the late-exposure group, tylosin treatment reduced total bacterial numbers only at day 10 (Fig. 3B; Table S1B), suggesting that exposure has less impact on the established microbiota. In the late-exposure group, tylosin did not affect *Snodgrassella* abundance (Fig. 4B; Table S1C) but did cause some changes in 16S rRNA copy numbers for other bacteria, such as

**(A)** Bacterial abundance in the **early** exposure group

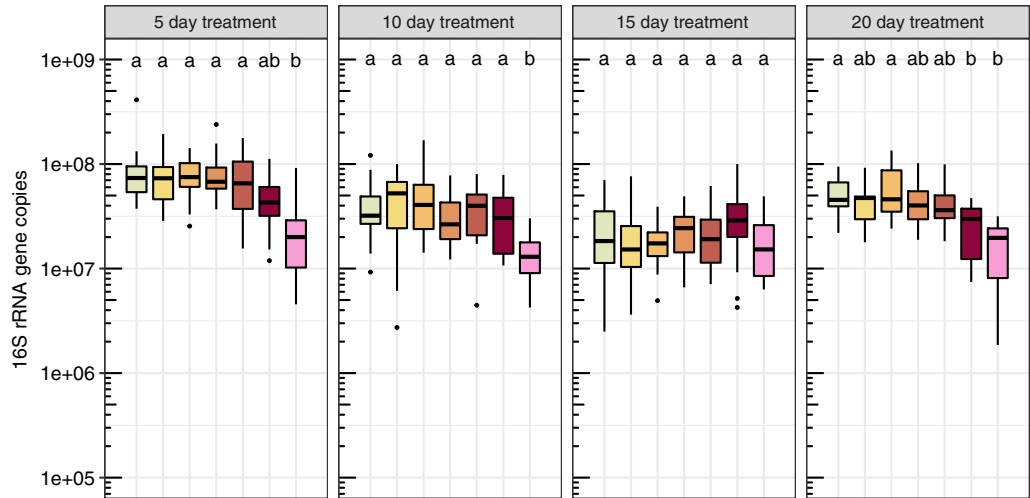

**(B)** Bacterial abundance in the **late** exposure group

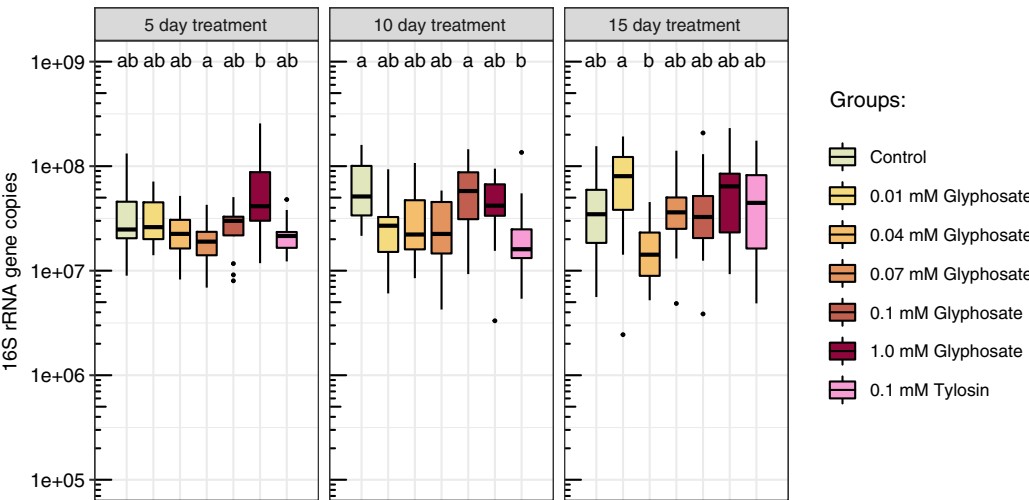

**FIG 3** Effects of different concentrations of glyphosate and of tylosin on bacterial abundance in the guts of laboratory-raised honey bees. (A) Bees at 1 day postemergence (the stage of acquiring the microbiota) were treated with glyphosate or tylosin for 20 days (early exposure). From left to right, box plots show the total 16S rRNA copy numbers in the guts of control bees, of 0.01, 0.04, 0.07, 0.1, and 1.0 mM glyphosate-fed bees, and of 0.1 mM tylosin-fed bees sampled at days 5 ($n = 15$ each), 10 ($n = 15$ each), 15 ($n = 15, 15, 15, 15, 15, 14,$ and 15), and 20 ($n = 10, 10, 11, 11, 11, 11,$ and 10) after the start of treatment. (B) Bees at 5 days postemergence (with established gut microbial communities) were treated with glyphosate or tylosin for 15 days (late exposure). From left to right, box plots show the total 16S rRNA copy numbers in the guts of control bees, of 0.01, 0.04, 0.07, 0.1, and 1.0 mM glyphosate-fed bees, and of 0.1 mM tylosin-fed bees sampled at days 5 ($n = 13, 13, 13, 13, 13, 13,$ and 12), 10 ($n = 12$ each), and 15 ($n = 12$ each) after the start of treatment. Groups with different letters are statistically significantly different ($P < 0.05$, Kruskal-Wallis test followed by Dunn's multiple-comparison test) (Table S1B).

reductions for *Lactobacillus* Firm-4 and *Lactobacillus* Firm-5 at specific sampling times and an increase for *Commensalibacter* at day 15 (Fig. S2; Table S1C).

Therefore, the higher concentrations of glyphosate impact the abundance of specific bacterial members, especially *Snodgrassella*, in both developing and established gut communities. In contrast, tylosin effects extend to other core bacterial members and appear to differ depending on whether exposure is early or late, with effects being greater for developing communities.

**Survivorship of worker honey bees exposed to glyphosate or tylosin.** The life spans of young worker bees chronically treated with different concentrations of glyphosate and with tylosin were monitored at two stages, during and after gut

**(A)** *Snodgrassella alvi* abundance in the **early** exposure group

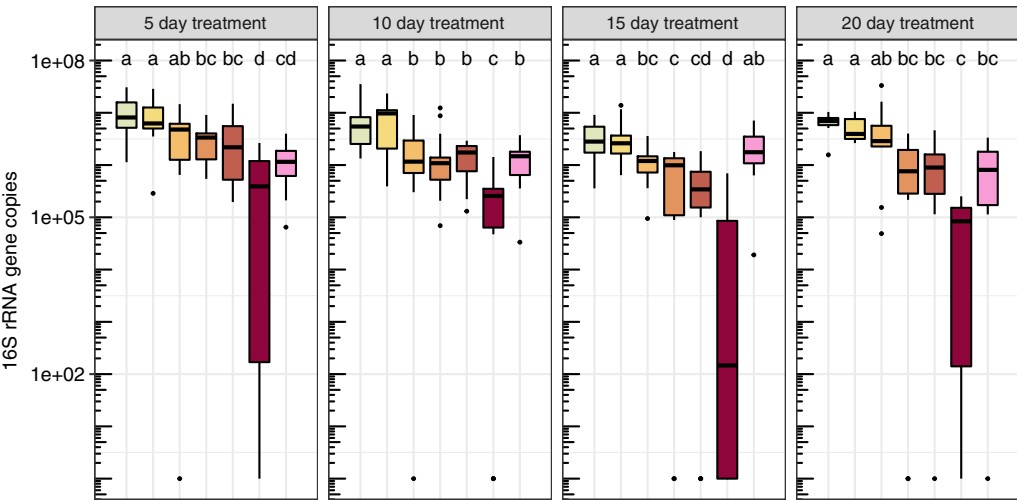

**(B)** *Snodgrassella alvi* abundance in the **late** exposure group

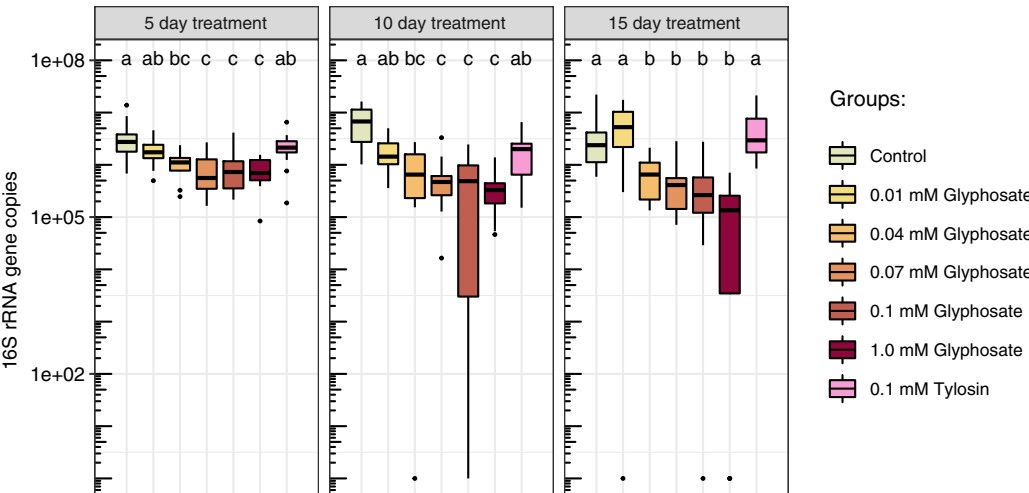

Groups:

- Control
- 0.01 mM Glyphosate
- 0.04 mM Glyphosate
- 0.07 mM Glyphosate
- 0.1 mM Glyphosate
- 1.0 mM Glyphosate
- 0.1 mM Tylosin

**FIG 4** Effects of different concentrations of glyphosate and of tylosin on *Snodgrassella* abundance in the guts of laboratory-raised honey bees. (A) Bees at 1 day postemergence were treated with glyphosate or tylosin for 20 days (early exposure). From left to right, box plots show *Snodgrassella* 16S rRNA copy numbers in the guts of control bees, of 0.01, 0.04, 0.07, 0.1, and 1.0 mM glyphosate-fed bees, and of 0.1 mM tylosin-fed bees sampled at days 5 ($n = 15$ each), 10 ($n = 15$ each), 15 ($n = 15$, 15, 15, 15, 15, 14, and 15), and 20 ($n = 10$, 10, 11, 11, 11, 11, and 10) after the start of treatment. (B) Bees at 5 days postemergence were treated with glyphosate or tylosin for 15 days (late exposure). From left to right, box plots show *Snodgrassella* 16S rRNA copy numbers in the guts of control bees, of 0.01, 0.04, 0.07, 0.1, and 1.0 mM glyphosate-fed bees, and of 0.1 mM tylosin-fed bees sampled at days 5 ($n = 13$, 13, 13, 13, 13, 13, and 12), 10 ($n = 12$ each), and 15 ($n = 12$ each) after the start of treatment. Groups with different letters are statistically significantly different ($P < 0.05$, Kruskal-Wallis test followed by Dunn's multiple-comparison test) (Table S1C).

microbiota acquisition, and during two seasons, fall 2018 and summer 2019. In the fall 2018 experiment, bees were also sampled every 5 days to assay changes in gut microbial abundance and composition.

In the fall 2018 experiment, 1-day-old bees treated with 0.1 mM or 1.0 mM glyphosate for 20 days (early exposure) exhibited increased mortality compared to control bees (Fig. 6A; Table S3). Similar effects were observed for 5-day-old bees treated with 0.1 mM or 1.0 mM glyphosate for 15 days (late exposure) (Fig. 6B; Table S3). Surprisingly, 5-day-old bees chronically treated with 0.01 mM glyphosate exhibited higher survival rates than control bees (Fig. 6B; Table S3). Treatment with 0.1 mM tylosin

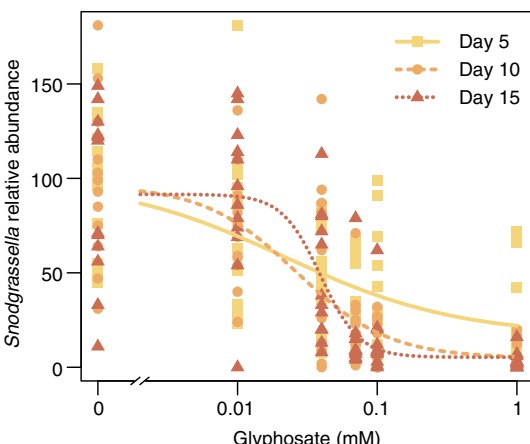

**FIG 5** Dose-response curves for *Snodgrassella* relative abundance in the guts of honey bees after treatment with different concentrations of glyphosate. (A) One-day-old adult worker bees at the stage of acquiring the microbiota were treated with glyphosate for 20 days (early exposure). (B) Five-day-old bees with established gut microbial communities were treated with glyphosate for 15 days (late exposure). Dots show *Snodgrassella* 16S rRNA copy numbers in the guts of sampled bees. For both panels, four-parameter log-logistic dose-response models were fitted using the functions drm and LL.4 (to fit and define the structure of the regression model) in the package drc (58) in R v3.5.2 (56) (Table S2).

increased mortality of bees only when exposure started immediately after emergence (Fig. 6A; Table S3).

In summer 2019, bees in the early-exposure group treated with 0.1 mM or 1.0 mM glyphosate exhibited increased mortality compared to control bees after 40 days of exposure (Fig. 6C; Table S3). This was also true for bees in the late-exposure group after 35 days of treatment (Fig. 6D; Table S3). However, we did not observe increased mortality of bees chronically treated with 0.1 mM tylosin (Fig. 6C and D; Table S3). Overall, bees in the late-exposure group experienced elevated mortality rates compared to bees in the early-exposure group, probably due to the extra handling and $CO_2$ exposure at day 5, when they were reassigned to new cup cages.

Potentially, the decreased survivorship of honey bees chronically exposed to 1.0 mM glyphosate is due to effects on bee physiology other than perturbations of the gut microbiota. To investigate this, we performed a laboratory experiment in fall of 2019 in which we exposed 1-day and 5-day-old bees to sterile sucrose syrup with and without 1.0 mM glyphosate throughout their life span (bees also had free access to sterile pollen). These bees were deprived of their normal microbiota, an abnormal state that is possible only under laboratory conditions. For both age groups, glyphosate-exposed bees exhibited higher mortality than control bees (Fig. S3; Table S3). Thus, the decreased survivorship of honey bees exposed to 1.0 mM glyphosate under laboratory conditions may involve effects other than those on the microbiome.

**Microbial diversity analysis.** To investigate potential changes in microbial community diversity between control and treatment groups, we used the relative abundance profiles obtained by 16S rRNA amplicon sequencing to estimate dissimilarity matrices using Bray-Curtis dissimilarity, which reflects community composition, and weighted UniFrac distance, which takes into account phylogenetic relationships among members of the bacterial communities and amplicon sequence variant (ASV) abundance (29). In addition, we calculated alpha diversity by means of Shannon's H index, a commonly used metric that accounts for both taxon richness and evenness of ASVs in each sample.

Based on pairwise permutational multivariate analysis of variance (PERMANOVA) using Bray-Curtis dissimilarity indices, and visualized using principal-coordinate analysis (PCoA), gut community compositions were usually significantly affected by glyphosate doses higher than 0.04 mM in the early-exposure group or 0.07 mM in the late-exposure

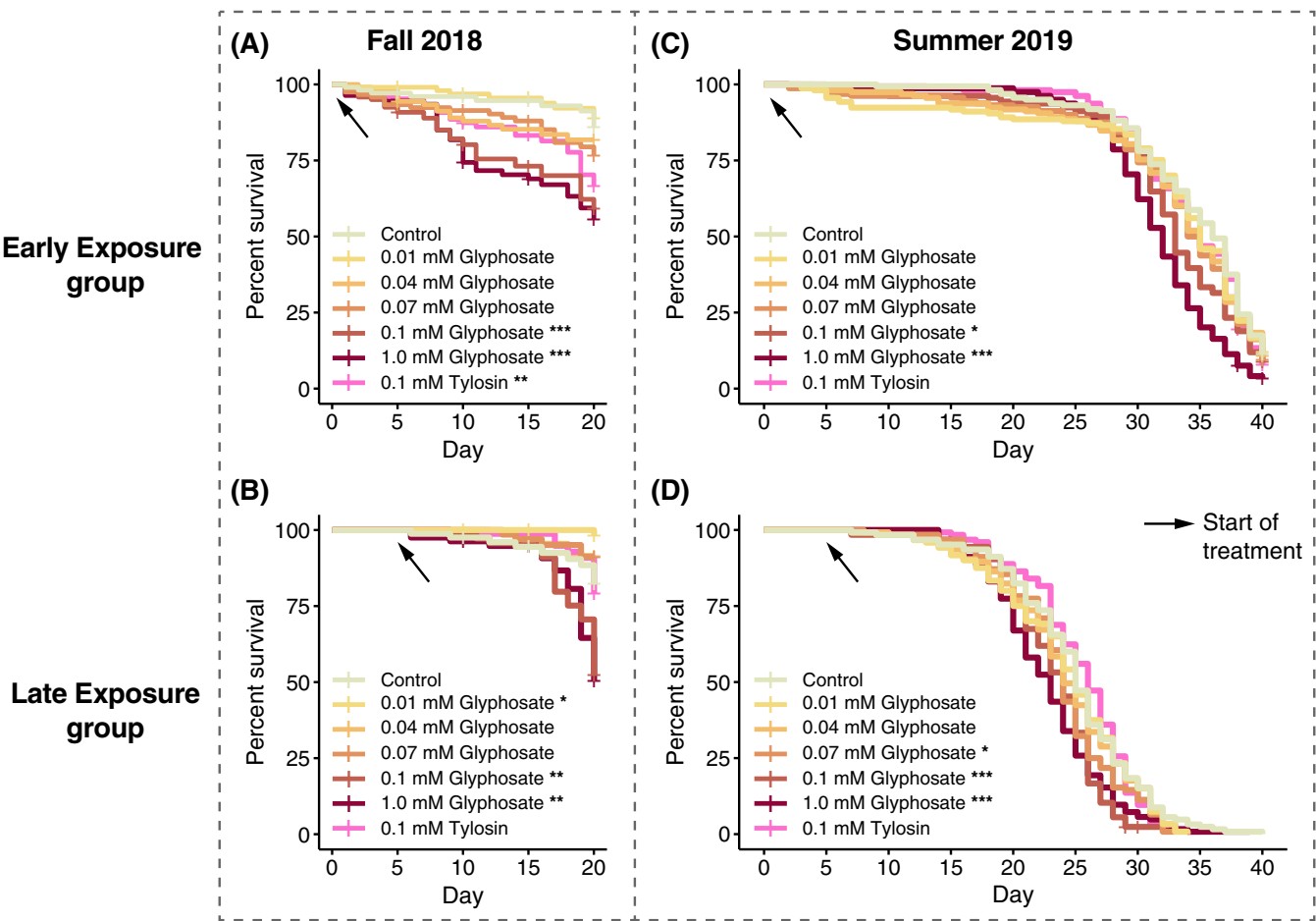

**FIG 6** Percent survival of age-controlled, caged bees treated with different doses of glyphosate or tylosin, shown as a Kaplan-Meier survival curve. (A and B) In fall of 2018, 1-day-old bees (A) and 5-day-old bees (B) were divided into seven groups, composed of 4 cup cages each with 26 to 30 bees, which were fed 0.01, 0.04, 0.07, 0.1, or 1.0 mM glyphosate in sterile sucrose syrup, 0.1 mM tylosin in sterile sucrose syrup, or sterile sucrose syrup for 20 and 15 days, respectively. Bees were sampled from each group every 5 days after the start of treatment, and dead bees were counted and removed from cup cages in a daily census. (C and D) The experiment was replicated in the summer of 2019, when 1-day-old bees (C) and 5-day-old bees (D) were divided into seven groups, composed of 4 cup cages each with 38 to 40 or 30 to 32 bees, respectively, which were fed 0.01, 0.04, 0.07, 0.1, or 1.0 mM glyphosate in sterile sucrose syrup, 0.1 mM tylosin in sterile sucrose syrup, or sterile sucrose syrup for 40 and 35 days, respectively. Bees were not sampled during treatment; dead bees were removed from cup cages in a daily census. The Cox proportional hazards model was implemented in the survival package in R (Table S3). *, $P < 0.05$; **, $P < 0.01$; ***, $P < 0.001$.

group (Fig. 7; Table S4). On the other hand, pairwise PERMANOVA using weighted UniFrac indices detected effects of only the higher dose of glyphosate (1.0 mM) and only after 10 days of exposure (Fig. S4; Table S4). Gut community compositions of tylosin-treated bees were significantly different from those of both control and glyphosate-treated bees throughout the experiments, regardless of exposure group, sampling time, and dissimilarity matrix used (Fig. 7, Fig. S4, and Table S4).

Regarding alpha diversity, Shannon's H index was significantly lower in tylosin-treated bees than in control bees at most sampling times in the early-exposure group but only at the last sampling time in the late-exposure group (Fig. S5; Table S5). No significant changes were observed between control and glyphosate-treated bees (Fig. S5; Table S5).

## DISCUSSION

Anthropogenic chemicals potentially affect gut microbiotas, with consequences for hosts themselves. Previous evidence of impacts of glyphosate, the main herbicide used globally in agriculture, on the honey bee gut microbiota prompted us to investigate the effects of dose and of timing of exposure. Honey bees acquire their gut microbiotas during the first few days after emergence from the pupal stage, and these symbionts

**(A)** PCoA of Bray-Curtis dissimilarity in the **early** exposure group

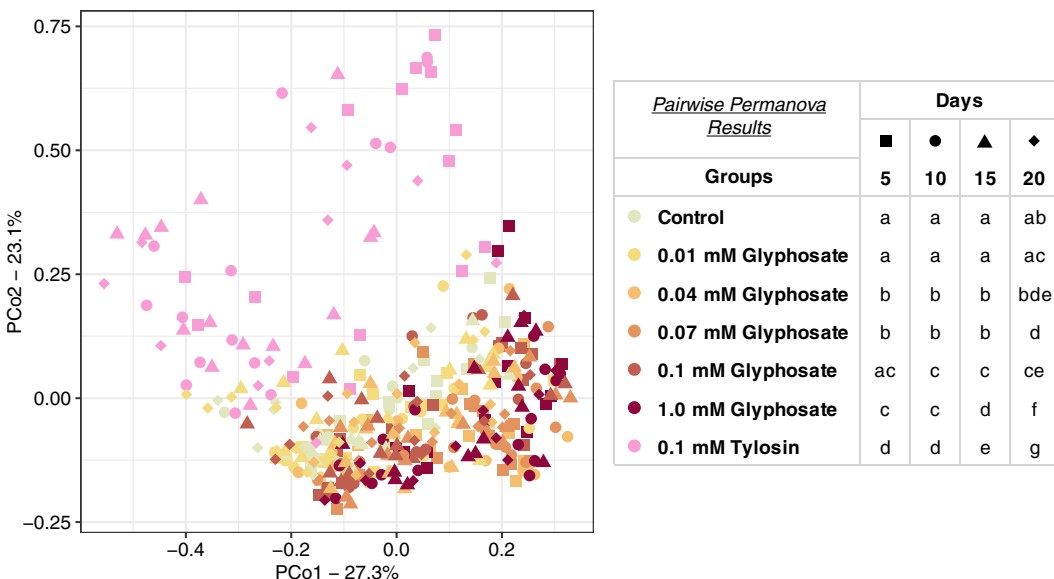

**(B)** PCoA of Bray-Curtis dissimilarity in the **late** exposure group

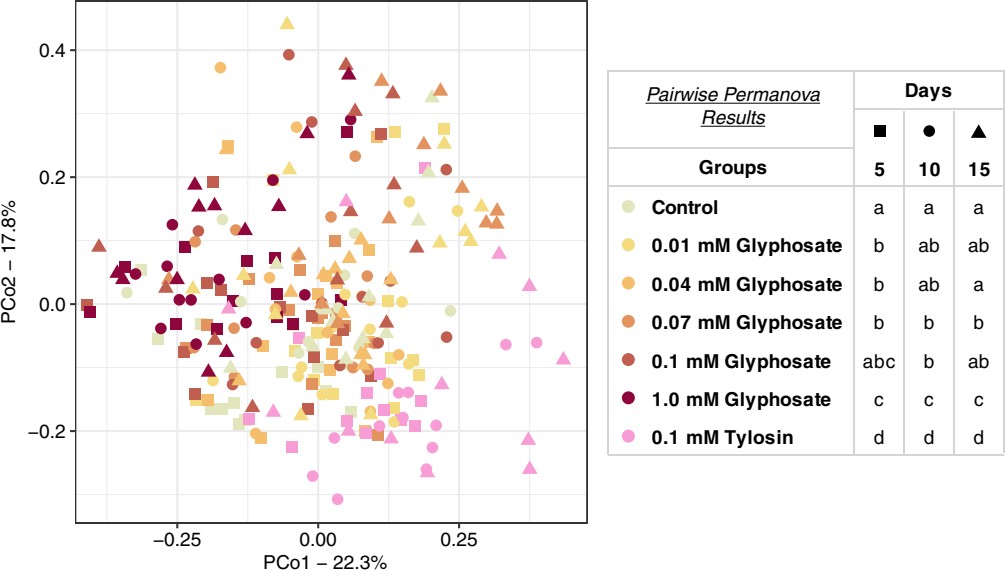

**FIG 7** Principal-coordinate plots of gut community compositions of bees treated with glyphosate or tylosin at different stages of gut microbial acquisition using Bray-Curtis dissimilarity. (A) Bees at 1 day postemergence were treated with glyphosate or tylosin for 20 days (early exposure). Control bees, 0.01, 0.04, 0.07, 0.1, and 1.0 mM glyphosate-fed bees, and 0.1 mM tylosin-fed bees were sampled at days 5 ($n = 15$ each), 10 ($n = 15$ each), 15 ($n = 15, 15, 15, 15, 15, 14$, and 15), and 20 ($n = 10, 10, 11, 11, 11, 11$, and 10). (B) Bees at 5 days postemergence were treated with glyphosate or tylosin for 15 days (late exposure). Control bees, 0.01, 0.04, 0.07, 0.1, and 1.0 mM glyphosate-fed bees, and 0.1 mM tylosin-fed bees were sampled at days 5 ($n = 13, 13, 13, 13, 13, 13$, and 12), 10 ($n = 12$ each), and 15 ($n = 12$ each) after the start of treatment. Within each sampling day, groups with different letters are statistically significantly different ($P < 0.05$, pairwise PERMANOVA with 999 permutations) (Table S4).

have multiple functions in bee development, behavior, nutrition, and immunity (2, 3, 5, 17, 19, 30, 31). Changes in microbial abundance or composition due to chemical exposure can interfere with these functions (9, 10, 13). Newly emerged worker bees perform crucial tasks in the hive, such as feeding and taking care of the brood (2), and they exhibit high physiological and behavioral plasticity (32), potentially making them more susceptible to perturbations of the gut microbiota. Exposure of young bees can

result from direct applications of chemicals in colonies or from residues in pollen and nectar that foragers bring to colonies. While some effects, such as those due to insecticide exposure, are well known (33–36), the effects of other agrochemicals, in particular antibiotics (10), fungicides (37), and herbicides (13), on the honey bee gut microbiota and health are starting to gain more attention.

Here, we demonstrate that laboratory-raised worker bees chronically treated with glyphosate have abnormal microbiotas. Bees, both at the stage of microbiota acquisition and with established microbiotas, were fed glyphosate in concentrations ranging from those detected in environmental samples (0.01 mM) (23–25) to those measured in nectar and pollen collected by foragers from plants recently sprayed with glyphosate (1.0 mM) (20). These concentrations are far lower than concentrations sprayed in the field from glyphosate-based formulations; they represent levels documented to occur in at least some circumstances (20, 23–25). Bees were sampled throughout the experiment to assess potential changes in microbial abundance and composition. Effects of glyphosate on the microbiota, especially on *Snodgrassella*, one of five main core bacterial species in the bee gut, were detected throughout the sampling times (Fig. 4). Although we expected that early exposure, during initial colonization by the microbiota, might have a greater impact, we found instead that impact was similar regardless of whether treatment started before or after microbiota acquisition. For both early and late exposure, these effects increased from lower to higher concentrations of glyphosate, exhibiting a dose-response relationship (Fig. 5). Because we sampled bees destructively, we could not follow changes within a single gut community over time, and this limitation was compounded by the large variation in community composition among individual bees. Despite variation, experiments of this study nonetheless revealed strong effects that are consistent with those of previous studies (11, 13).

The species consistently most affected by glyphosate was *Snodgrassella alvi*, a Gram-negative bacterium that primarily colonizes the hindgut, where it forms a biofilm on the ileum wall. This biofilm may facilitate colonization by other beneficial bacteria and/or act as a mechanical barrier against pathogen invasion (2). Previous findings have suggested that *Snodgrassella* strains vary in susceptibility to glyphosate. Motta et al. (13) reported two strains of *Snodgrassella* that were tolerant to glyphosate when cultured *in vitro*, whereas all the other strains tested were susceptible. However, the overall abundance of *Snodgrassella* decreased in every experiment in which honey bees were exposed to sufficient concentrations of glyphosate, as shown in this study and others (11, 13). Thus, tolerant *Snodgrassella* strains either are atypical or are not resistant in the context of an intact gut community.

Previous studies have shown that sublethal doses of glyphosate affect the microbiotas of young and adult worker bees under laboratory conditions, but no dose-response relationship was determined. Dai et al. (12) studied the midgut microbial composition of newly emerged bees exposed to glyphosate (0.8, 4, and 20 mg/liter) as larvae. They found changes in microbial diversity between control and treatment bees, especially in bees fed 20 mg/liter glyphosate at the larval stage. Besides the effects on the microbiota, they observed lower survival rates for larvae fed 4.0 and 20 mg/liter glyphosate. In addition, we previously found that recently emerged bees treated with glyphosate (16.91 and 169.1 mg/liter) have altered hindgut microbiota with reduced populations of beneficial bacteria (13). We also detected reduced microbial abundance in adult workers released in their hives after being fed 5 or 10 mg/liter glyphosate (13). Blot et al. (11) found that glyphosate exposure at 253.6 or 1,268.2 mg/liter affects the gut microbiota of adult worker bees under laboratory conditions. Together, these results corroborate the findings described in this study, in which microbiotas of young bees exposed to glyphosate were affected, both during and after establishment of the gut community.

We compared effects of glyphosate to effects of tylosin, an antibiotic commonly used in beekeeping for the control of American and European foulbrood disease (38–40). Tylosin mainly affected total bacterial abundance and *Snodgrassella* abundance when treatment started immediately following worker emergence, unlike

mSystems®

glyphosate, which decreased *Snodgrassella* abundance regardless of treatment timing. Tylosin strongly reduced the abundance of Gram-positive bacteria, which was not the case for glyphosate. This corroborates previous findings on the effects of other antibiotics, such as tetracycline (10, 41) and penicillin-streptomycin (9), on the bee gut microbiota. The lack of effects of glyphosate on Gram-positive bacteria in the bee gut under laboratory conditions, especially on *Lactobacillus*, may be explained by the fact that these bacteria lack a functional shikimate pathway, which contains the enzyme EPSPS, the target of glyphosate (13). However, occasional decreases in abundance for these bacteria may reflect their reliance on the uptake of essential nutrients produced by glyphosate-affected bacteria. Because essential amino acids are absorbed by the host midgut and low in the hindgut (18), the hindgut community must synthesize these, using ammonia from host waste that enters at the midgut-hindgut juncture. Thus, some species rely on cross-feeding, at least for amino acids, and possibly for other nutrients, and disruption of glyphosate-sensitive species can have ripple effects throughout the community.

A critical question is how observed shifts in microbiota affect bee survivorship. Our survivorship experiments used bees caged in the laboratory (Fig. 6) and thus failed to capture sources of mortality that affect bees in colonies. Indeed, previous experiments with a single glyphosate dose showed that exposure lowered the ability of caged bees to survive following challenge with pathogenic *Serratia marcescens* but did not affect survivorship otherwise (13). Similarly, tetracycline exposure increased mortality of bees in colonies and of caged laboratory bees challenged with *S. marcescens* but did not affect survivorship of caged control bees (10). We found that, under laboratory conditions and without pathogen challenge, only the higher doses of glyphosate (0.1 mM and 1.0 mM) reduced the life span of young worker bees. In studies that did not examine the role of gut microbiota, glyphosate was shown to directly compromise larval development and adult behavior (12, 42–45), although some other studies failed to detect effects on survivorship (20, 46, 47). Tylosin, which had large effects on the gut microbiota, increased mortality in only one of the four trials with caged laboratory bees (Fig. 6). Further experiments tracking exposed bees in field colonies could determine whether the observed microbial perturbations are detrimental for bees in the hive environment, where they are more exposed to opportunistic pathogens and other environmental stressors.

The higher doses of glyphosate used in our study are similar to the ones quantified in nectar and pollen from plants recently treated with glyphosate-based formulations (20). Although foraging bees may encounter glyphosate at these concentrations, glyphosate-susceptible plants die a few days after exposure and glyphosate concentrations decrease with time on those plants (20). Therefore, it is unlikely that young worker bees would be chronically exposed to these concentrations of glyphosate. The lower dose of glyphosate tested, 0.01 mM (1.691 ppm), did not affect the gut microbiota or the survival rates of honey bees (except in one trial in which survivorship of control bees was lower). This lower dose resembles concentrations previously measured in the environment in water sources (23–25) and in nectar samples taken from colonies in a semi-field experiment (20) but is still higher than concentrations documented in honey, the most obvious source of chronic exposure (21, 22, 48, 49).

Another issue in laboratory experiments is that captive honey bees usually do not defecate, causing DNA from dead bacterial cells to accumulate within the gut. Therefore, DNA samples used to profile microbial communities may give a biased picture of abundances of living bacterial cells. Indeed, for soil microbial communities, studies have demonstrated that relic DNA can obscure estimates of microbial diversity (50). Most experiments documenting effects of glyphosate on the bee gut microbiota have been performed under laboratory conditions using DNA extracted from dissected guts. All such experiments have revealed a strong reduction in the abundance of *Snodgrassella*, regardless of potential issues with dead bacterial cells (11, 13). To reduce artifacts resulted from relic DNA, we extracted RNA and compared relative rRNA abundances; the effects of glyphosate were similar to those observed in studies based on genomic

DNA. However, this may not be true for all chemical stressors. In a previous study, we extracted both DNA and RNA from individual bee guts, and only RNA samples showed significant effects of tylosin treatment for some bacterial taxa (13). Therefore, the use of RNA samples, or removing relic DNA before DNA extraction from viable cells, could provide more accurate measures of the perturbations to living bacterial populations.

While glyphosate exposure had substantial and consistent effects on abundance of individual bacterial taxa, these were not reflected in Shannon's H estimates of alpha diversity (Fig. S5; Table S5) or in total bacterial abundance (Fig. 3). Thus, standard measures of diversity fail to capture changes in community composition that have large functional consequences, given that individual members of the community have very different metabolic capabilities and occupy different locations in the gut (13, 14). We also note that the diversity measures in our study were based on profiles of amplicons within 16S rRNA genes and thus would not reveal changes in strain-level composition. Potentially, using more variable marker loci would reveal reductions in strain diversity, as observed in a study of community shifts following tetracycline exposure (41). For example, a few strains of *Snodgrassella* appear to be glyphosate resistant (13), and these might increase in frequency following exposure.

In summary, glyphosate concentrations ranging from 0.07 mM to 1.0 mM reduced the abundance of beneficial bee gut bacteria, particularly *Snodgrassella*, in a dose-dependent way. Such effects occurred regardless of the stage of microbiota establishment and regardless of duration of exposure. Although the concentrations used in this study are higher than the ones commonly detected in honey samples, they are in the range of concentrations found in nectar and pollen of recently exposed plants that may be used as a source of food by foraging bees. These concentrations are far lower than the concentrations sprayed in fields, where foraging bees can be topically exposed. Although a chronic exposure would be unlikely in this last case, direct spraying on foragers is a potential route through which hives might be contaminated.

## MATERIALS AND METHODS

**Chemicals and solutions.** Glyphosate standard was purchased from Research Products International, USA (lot 32612-38399). Tylosin tartrate was purchased from GoldBio, USA (lot 2313.081915A). For all the experiments, glyphosate and tylosin were initially dissolved in distilled water and then diluted to the final concentration with filter-sterilized 0.5 M sucrose syrup. Glyphosate concentrations were expressed in molarity (millimolar units) but can be converted to milligrams per liter by multiplying by the molecular weight of glyphosate, which is 169.1 g/mol, or to milligrams per kilogram by multiplying by the molecular weight of glyphosate and then dividing by the density of sucrose syrup, considered here as 1.23 g/ml. For example, the lower concentration of glyphosate used in our experiments was 0.01 mM, which is 1.691 (0.01 × 169.1) mg/liter, or 1.375 (1.691/1.23) mg/kg. The same can be done for tylosin, whose molecular weight is 916.1 g/mol.

**First chronic exposure of honey bees to glyphosate and tylosin.** Approximately 2,000 late-stage pupae, with pigmented eyes but lacking movement, were removed from two brood frames from a hive of the European *Apis mellifera* kept at the University of Texas–Austin in fall of 2018. Pupae were transferred to clean plastic bins and placed in an incubator at 35°C and ~60% relative humidity to simulate hive conditions until they emerged as adults. Healthy newly emerged workers (NEWs) were transferred in groups of 26 to 30 to cup cages containing sterile sucrose syrup and bee bread (total of 56 cup cages). The bee bread was mixed with a bee gut homogenate so the bees could acquire their normal microbiota. In brief, guts were aseptically pulled out from healthy workers from the same hive and homogenized with equal proportions of 1× phosphate-buffered saline (PBS) and sterile sucrose syrup. Gut homogenate (200 $\mu$l) was transferred to the bee bread provided to the bees in each cup cage. Then, bees were divided into two main groups.

**(i) Early-exposure group.** Right after the gut homogenate treatment, 28 cup cages were divided into 7 subgroups which were fed 0.01, 0.04, 0.07, 0.1, or 1.0 mM glyphosate in sterile sucrose syrup, 0.1 mM tylosin in sterile sucrose syrup, or sterile sucrose syrup for 20 days. Fifteen bees were sampled from each group at days 5, 10, 15, and 20 after the start of the chemical feeding, placed in 5-ml Falcon tubes, and stored at −80°C. Bees were quickly immobilized with $CO_2$ before each sampling. Survival rates were monitored, and dead bees were removed in a daily census.

**(ii) Late-exposure group.** After 5 days of gut homogenate treatment, which is sufficient time for establishment of the gut microbiota (51), the remaining cup cages were divided into 7 subgroups which were fed similarly to the early-exposure group. Fifteen bees were sampled from each group at days 5, 10, and 15 after the start of the chemical feeding, placed in 5-ml Falcon tubes, and stored at −80°C. Bees were quickly immobilized with $CO_2$ before each sampling. Survival rates were also monitored, and dead bees were removed in a daily census.

**Second chronic exposure of honey bees to glyphosate and tylosin.** Since honey bees were sampled throughout the first chronic exposure experiment, which required quick immobilization of bees with $CO_2$ every 5 days, we decided to repeat this experiment in summer 2019 without introducing potential side effects of sampling and $CO_2$ exposure. In this case, we collected two frames from a second hive, and pupae were allowed to emerge naturally under laboratory conditions (35°C and ~60% relative humidity). After 1 day, approximately 2,000 NEWs were transferred in groups of 40 to cup cages containing sterile sucrose syrup and bee bread mixed with a gut homogenate (total of 50 cup cages). Then, bees were divided into two main groups as for the first chronic exposure experiment.

**(i) Early-exposure group.** Right after the gut homogenate treatment, 28 cup cages, containing 1,120 NEWs, were divided into 7 subgroups which were fed 0.01, 0.04, 0.07, 0.1, or 1.0 mM glyphosate in sterile sucrose syrup, 0.1 mM tylosin in sterile sucrose syrup, or sterile sucrose syrup for 40 days. Each subgroup consisted of 4 cup cages each containing 38 to 40 bees. Fresh treatments were added in the middle of the experiment. Survival rates were monitored for 40 days, and dead bees were removed in a daily census.

**(ii) Late-exposure group.** After 5 days of gut homogenate treatment, bees from the remaining 22 cup cages, containing 870 young workers in total, were briefly immobilized with $CO_2$, combined, and then randomly transferred in groups of 30 to 32 bees to new cup cages, totaling 28 cup cages. Cup cages were divided into 7 subgroups which were fed similarly to the early-exposure group. Survival rates were monitored for 35 days, and dead bees were removed in a daily census. Compared to bees in the early-exposure group, bees in the late-exposure group experienced an overall increase in mortality, probably due to the extra $CO_2$ exposure and handling.

**Chronic exposure of microbiota-depleted honey bees to glyphosate.** In fall of 2019, approximately 600 late-stage pupae were removed from a brood frame from the hive used in the second chronic-exposure experiment, transferred to clean plastic bins, and placed in an incubator at 35°C and ~60% relative humidity until emerging as adults. Healthy NEWs were transferred to cup cages containing sterile sucrose syrup and sterile bee bread (total of 16 cup cages each with 25 to 30 bees). No gut homogenate was added to the bee bread, and bees remained microbiota free throughout the experiment. This time, the early-exposure group consisted of 8 cup cages containing 1-day-old microbiota-depleted bees, whereas the late-exposure group consisted of 8 cup cages containing 5-day-old microbiota-depleted bees. Each group was divided into 2 subgroups which were fed sterile sucrose syrup with or without 1.0 mM glyphosate for 40 days. Survival rates were monitored, and dead bees were removed in a daily census.

**Gut dissection and RNA extractions.** Bees were removed from the −80°C freezer, and guts were dissected with flame-sterilized forceps under aseptic conditions and on ice. Total RNA was extracted from each dissected gut using the Quick-RNA miniprep kit (Zymo Research). In brief, guts were removed from bee abdomens and crushed in 100 $\mu$l of RNA lysis buffer, resuspended in a total of 600 $\mu$l of the same solution, and transferred to a capped vial containing 0.5 ml of 0.1-mm zirconia beads (BioSpec Products Inc.). Samples were subjected to bead beating twice for 30 s each time, centrifuged at 14,000 rpm for 30 s, and transferred to a new microtube. After this step, extraction followed the protocol provided by Zymo Research. Final RNA samples were dissolved in 50 $\mu$l of water and stored at −80°C. cDNA was synthesized from 2 $\mu$l of RNA sample using the qScript cDNA synthesis kit (Quantabio, Beverly, MA, USA) following the manufacturer's instructions, and stored at −20°C.

**Quantitative PCR.** cDNA samples were diluted 10-fold to be used as the template for qPCR analyses. The universal bacterial 16S rRNA gene primers 27F (5′-AGAGTTTGATCCTGGCTCAG-3′) and 355R (5′-CT GCTGCCTCCCGTAGGAGT-3′) were used to amplify total copies of 16S rRNA in each sample on an Eppendorf Mastercycler ep realplex instrument. Duplicate 10-$\mu$l reactions were carried out with 5 $\mu$l Kapa SYBR fast 2× master mix (Kapa Biosystems), 0.05 $\mu$l of each primer at 100 $\mu$M, 3.9 $\mu$l $H_2O$, and 1.0 $\mu$l template DNA. The cycling conditions consisted of 95°C for 3 min followed by 5 cycles of a three-step PCR (95°C for 5 s, 65 to 60°C for 15 s [a decrease of 1°C per cycle], and 68°C for 20 s) and 35 cycles of a second three-step PCR (95°C for 5 s, 60°C for 15 s, and 68°C for 20 s). Quantification was based on standard curves from amplification of the cloned target sequence in a pGEM-T vector (Promega).

**16S rRNA library preparation and sequencing.** Library preparation consisted of two PCR steps. PCR 1 was designed to amplify the V4 region of the 16S small-subunit (SSU) rRNA gene and was performed in 20-$\mu$l triplicate reactions using primers 515F (5′-TCGTCGGCAGCGTCAGATGTGTATAAGAGACAGGTGY CAGCMGCCGCGGTA-3′) and 806R (5′-GTCTCGTGGGCTCGGAGATGTGTATAAGAGACAGGGACTACHVGGG TWTCTAAT-3′) (both at 200 nM final concentration), 8 $\mu$l of 5PRIME HotMasterMix (2.5×; Quantabio, Beverly, MA, USA), and 1 $\mu$l of template DNA. Cycling conditions were as follows: 94°C for 3 min; 30 cycles of 94°C for 45 s, 50°C for 60 s, 72°C for 90 s; then 72°C for 10 min. PCR 1 products were combined, purified with 0.8× HighPrep PCR magnetic beads (MagBio, Gaithersburg, MD, USA), and diluted to a final volume of 52.5 $\mu$l.

PCR 2 was designed to attach dual indices and an Illumina sequencing adapter to the PCR 1 product and was performed in 25-$\mu$l single reactions using a unique combination of index primers N7XX (5′-CAAGCAGAAGACGGCATACGAGATNNNNNNGTCTCGTGGGCTCGG-3′) and S5XX (5′-AATGATACGGCG ACCACCGAGATCTACACNNNNNNTCGTCGGCAGCGTC-3′) (both at 400 nM final concentration), 10 $\mu$l of 5PRIME HotMasterMix (2.5×, Quantabio, Beverly, MA, USA), and 5 $\mu$l of PCR 1 product. Cycling conditions were as follows: 94°C for 3 min; 10 cycles of 94°C for 20 s, 55°C for 15 s, 72°C for 60 s; then 72°C for 10 min. PCR 2 products were purified with 0.8× HighPrep PCR magnetic beads (MagBio, Gaithersburg, MD, USA), diluted to a final volume of 27.5 $\mu$l, and quantified fluorometrically (Qubit. Thermo Fisher Scientific Inc.). Samples (50 ng each) were split into two pooled libraries. The first pooled library consisted of samples from the early-exposure group (1-day-old bees treated with chemicals for 20 days; total of 389 samples).

The second pooled library consisted of samples from the late-exposure group (5-day-old bees treated with chemicals for 15 days; total of 258 samples). Each library was loaded onto an Illumina iSeq cartridge according to the manufacturer's instructions and subjected to Illumina sequencing on the iSeq platform (2 × 150 sequencing run; instrument model number FS10000184). PhiX (5%) was used to increase library diversity.

**Processing of 16S rRNA amplicon data.** Illumina sequence reads were demultiplexed on the basis of the barcode sequences by the iSeq software, then processed according to exposure group in QIIME 2 version 2019.10 (52). Due to the lack of sufficient overlap between forward and reverse reads, downstream analyses were performed with forward reads only. Primer sequences were removed using the cutadapt plugin (53). Then, reads were truncated to a length of 120, filtered, and denoised, and chimeric reads were removed using the DADA2 plugin (54). Taxonomy was assigned to amplicon sequence variants (ASVs) using the SILVA database in the feature-classifier plugin (55). Reads with less than 0.1% abundance were removed from the data set using the feature-table plugin, as were unassigned, mitochondrial, and chloroplast reads using the taxa filter-table plugin. Microbial diversity analyses were performed using the diversity plugin with a sampling depth of 1,000 reads per sample.

**Statistical analyses.** Statistical analyses were performed in R version 3.5.2 (56) or QIIME 2 version 2019.10 (52). Principal-component analyses (PCAs) were performed with the prcomp function in the R package stats to determine the similarity of the bacterial communities between control and treatment groups. For that, the taxonomic classification based on the SILVA database was complemented by assigning the ASVs to the major bacterial taxa of the bee gut microbiota based on a BLAST search against the NCBI nucleotide database. Then, the relative abundances of the phylotypes were transformed by using $\log_{10}(x + 1)$ and used to plot the PCAs. Comparisons of changes in bacterial abundance or alpha diversity (Shannon's H index) between control and treatment groups were performed using the non-parametric Kruskal-Wallis test followed by Dunn's multiple-comparison test, if significant, in R. Principal-coordinate analyses based on Bray-Curtis dissimilarity or weighted UniFrac were plotted using the R package phyloseq (57), and statistical tests were performed using pairwise permutational multivariate analysis of variance (PERMANOVA) tests with 999 permutations in QIIME 2. Dose-response models were fitted using the drm and LL.4 functions to fit and define the structure of the regression model, and the modelFit function was used to obtain a lack-of-fit test, all in the R package drc (58). Comparisons of survival rates between control and treatment groups were performed using Kaplan-Meier survival curves and the Cox proportional hazards model implemented in the R package survival (59).

**Data availability.** All sequence data are available in NCBI BioProject, no. PRJNA578403. The other data generated during this study are included in this article and the supplemental material.

## SUPPLEMENTAL MATERIAL

Supplemental material is available online only.

**FIG S1**, EPS file, 0.9 MB.

**FIG S2**, EPS file, 1 MB.

**FIG S3**, EPS file, 0.3 MB.

**FIG S4**, EPS file, 0.5 MB.

**FIG S5**, EPS file, 0.3 MB.

**TABLE S1**, XLSX file, 0.2 MB.

**TABLE S2**, XLSX file, 0.01 MB.

**TABLE S3**, XLSX file, 0.01 MB.

**TABLE S4**, XLSX file, 0.03 MB.

**TABLE S5**, XLSX file, 0.02 MB.

## ACKNOWLEDGMENTS

We thank the Moran and Ochman labs, especially Eli Powell for technical assistance, and Kim Hammond for maintaining the hives.

We declare that we have no competing interests.

Funding was provided by Coordenação de Aperfeiçoamento de Pessoal de Nível Superior, Brazil (13578-13-8), and the Donald D. Harrington Fellows Program, University of Texas–Austin, to E.V.S.M. and the USDA National Institute of Food and Agriculture (2018-67013-27540) to N.A.M.

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
