## [Reviewer comments · mSystems]

Impact of glyphosate on the honey bee gut microbiota: effects of intensity, duration and timing of exposure

Erick Motta and Nancy Moran

Corresponding Author(s): Erick Motta, University of Texas at Austin

Review Timeline:

Submission Date:	March 30, 2020
Editorial Decision:	June 1, 2020
Revision Received:	June 24, 2020
Accepted:	July 8, 2020

Editor: Sarah Hird

Reviewer(s): The reviewers have opted to remain anonymous.

Transaction Report:

DOI: <https://doi.org/10.1128/mSystems.00268-20>

June 1, 2020

Dr. Erick V. S. Motta
University of Texas at Austin
Integrative Biology
2506 Speedway
NMS 4.126
Austin, TX 78712

Re: mSystems00268-20 (Impact of glyphosate on the honey bee gut microbiota: effects of intensity, duration and timing of exposure)

Dear Dr. Erick V. S. Motta:

Thank you for your patience with our review of this manuscript during this chaotic time. In conjunction with the previous reviews, I had one additional reviewer look it over and I agree with them that this is a well written paper that, with a few minor edits, is suitable for publication in mSystems. Please make particular note of the comments about the statistical results.

Below you will find the comments of the reviewers.

To submit your modified manuscript, log onto the eJP submission site at <https://msystems.msubmit.net/cgi-bin/main.plex>. If you cannot remember your password, click the "Can't remember your password?" link and follow the instructions on the screen. Go to Author Tasks and click the appropriate manuscript title to begin the resubmission process. The information that you entered when you first submitted the paper will be displayed. Please update the information as necessary. Provide (1) point-by-point responses to the issues raised by the reviewers as file type "Response to Reviewers," not in your cover letter, and (2) a PDF file that indicates the changes from the original submission (by highlighting or underlining the changes) as file type "Marked Up Manuscript - For Review Only."

Due to the SARS-CoV-2 pandemic, our typical 60 day deadline for revisions will not be applied. I hope that you will be able to submit a revised manuscript soon, but want to reassure you that the journal will be flexible in terms of timing, particularly if experimental revisions are needed. When you are ready to resubmit, please know that our staff and Editors are working remotely and handling submissions without delay. If you do not wish to modify the manuscript and prefer to submit it to another journal, please notify me of your decision immediately so that the manuscript may be formally withdrawn from consideration by mSystems.

To avoid unnecessary delay in publication should your modified manuscript be accepted, it is important that all elements you upload meet the technical requirements for production. I strongly recommend that you check your digital images using the Rapid Inspector tool at <http://rapidinspector.cadmus.com/RapidInspector/zmw/>.

If your manuscript is accepted for publication, you will be contacted separately about payment when the proofs are issued; please follow the instructions in that e-mail. Arrangements for payment must be made before your article is published. For a complete list of **Publication Fees**, including

supplemental material costs, please visit our website.

Sincerely,

Sarah Hird

Editor, mSystems

Journals Department
Reviewer comments:

Reviewer #1 (Comments for the Author):

I commend the authors on a very well planned and executed set of experiments. My comments are uploaded in a word document.

General Comments to the Authors:

In this revised manuscript, Motta & Moran expand on a previous finding that glyphosate can impact honey bee gut microbiota by exploring dose-dependent effects of glyphosate on honey bee gut microbiota. The authors did this by chronically exposing bees to varying concentrations of glyphosate-containing sucrose solution. In addition, they compare the impacts of these treatment groups to a single concentration of a commonly used honey bee antibiotic, tylosin, apparently as a form of positive control. In general, they do find dose-dependent effects of glyphosate exposure on honey bee gut microbiota, although the effects vary greatly with sampling day (5-20), time period (Fall 2018 vs Summer 2019), and the component of the microbial community measured (abundance, diversity, and individual taxonomic units). Still, overall their results do suggest that increasing exposure to glyphosate likely has increasingly direct impacts on honey bee gut microbiota, and potentially indirect effects on survivorship. Although, the latter is speculative, as in a separate experiment they confirm that glyphosate has direct effects on survivorship when applied to bees without established gut microbes, thus they were unable to attribute decreased survivorship to changes in gut microbiotas per se.

I believe this is the second round of revisions based on previous comments, although I did not see an original manuscript file uploaded so am therefore limited in my ability to respond to the suggested revisions by Reviewer #2. I therefore attempt to synthesize the former reviews with my own and present my assessment below. In general, the referenced page numbers in the 'Response to Reviewers Comments' do not align with either 'Manuscript Text File' or 'Marked up Manuscript' which were uploaded for review. My line comments below therefore follow the uploaded versions. I believe for the most part, the cited line revisions do correspond to the highlighted sections in the 'Marked Up Manuscript' document, just not the correct lines. While annoying, I found most responses satisfactory. In general the manuscript was well written, but there were instances where awkward paragraphs lingered or where minor edits were needed.

Overall, I agree with the authors that this manuscript will be of much more interest to a broader audience than Reviewer #2 suggested. The authors give many stats from their previous research to back up this point. Moreover, while the target organism of study (honey bees) does have a devoted audience, the broader theory/context behind direct and indirect dose-dependent effects of pesticides on non-target organisms is quite ubiquitous

While the focus of the manuscript is on dose-depend responses, I found it interesting that there was less focus on the implications of their methodological approach, namely that both dose and the timing of exposure to glyphosate, have interactive effects on honey bee gut microbiota and survivorship. Although there was tremendous variation in microbiota response, I think this could use a sentence or two in the discussion

Figure 6 C vs D. Why did late exposure bees in the summer of 2019 experience ~ 10 day shorter survivorship overall? It appears from the points of inflection, that overall the experimental replicates experienced the most rapid mortality around 35 days in the early exposure,

compared to 25 days in the late exposure. This is a huge difference; was there an explanation I missed?

One big concern is the lack of reported statistical test results. There are several instances where statistical tests were performed but I was unable to find sufficient results reported in the text of the manuscript or the supplementary material: for example, references to significant p-values for the Kruskal-Wallis tests and Cox Proportional Hazard Models, but no reported Chi-square values, DF, parameter estimates or coefficients. I would highly recommend the authors to verify these exist and are reported appropriately.

Example: On line 535-538: Comparisons of bacterial abundance or alpha diversity were done using Kruskal-Wallis test and Dunn's multiple comparisons. However, no chi-square results are reported that I can find, either in the manuscript or in the supplementary tables. For example, Figures 3 and 4 show Total and *Snodgrassella* abundances, respectively, and have letters assigned from the post hoc test, but no chi square. Where are the Kruskal-Wallis tests reported? Shouldn't there be a chi-square value, df, and actual p-values reported? Without them reported I can't in faith accept the post hoc tests.

Line Comments:

Line 36. I suggest replacing 'their' with 'honey bee.' Also is 'microbiota' singular or plural? If plural, change to 'microbiota are' throughout. If singular, leave as is.

Line 88-92. A 3-sentence introductory paragraph is awkward and detracts from the overall effectiveness of the introduction. Perhaps combine with paragraph 2 about bees?

Line 119. Is there evidence of bacterial death in the hindgut due to changes in midgut microbiota? If so references? If not, why?

Line 173-174. Interesting, could this suggested increased secondary infection, or just that bacteria are passing through the gut? Should mention why this is important somewhere.

Line 220. Is it standard to refer to a Cox Proportional Hazards Model as 'coxph' in the reported stats? I see it is the command for the model in R, but I believe you should list out the actual test here, not the R command. Also, where are the other model parameter estimates and/or

coefficients? Are the reported values only for the significant treatments? There should be p-values and/or coefficients for all reported somewhere. Perhaps a missing Supplementary Table?

Line 247. Is OTA acceptable without explanation here? I don't think it was ever defined, but perhaps it was not necessary for the journal?

Line 253. I agree with Reviewer 2 that traditionally, a PERMANOVA of the centroids is often visualized by ellipses. My bigger concern is that the PERMANOVA is not reported, only a p-value. Where are the other parameters: df, ss, F-stats, etc.?

Line 263. Where are the Chi-squares reported? I only see the letters from the post hoc tests.

Line 349. I find these potential bacterial interactions very interesting and think it could be worthwhile to explain in a bit more detail.

Line 351-367. This paragraph reads like a list of studies. Maybe revise to link them together for flow? I would suggest focusing on a topic and summary sentence for the paragraph, and selecting the studies that are most relevant to the subject (impacts of glyphosate on survivorship).

Line 366. There so far has been little to no discussion of synergistic effects from other chemicals. I'd suggest removing this as it doesn't seem relative.

Line 368-371. This 2-sentence paragraph is awkward and seems out of place. Maybe combine with following paragraph about caveats?

Line 390. Did you really test 'acute' vs 'chronic'? Weren't all treatments chronic?

Line 435. So were there differences in mortality between the two time periods?

Line 545. I think you should explicitly call this a Cox Proportional Hazards Model.

Line 595. I think this reference to the Kruskal-Wallis test is problematic. You should have one Kruskal-Wallis test reported, if significant, then followed up with the post hoc Dunn's multiple comparisons. Where is the Kruskal-Wallis reported?

Responses to the editor and reviewer (mSystems00268-20)

For the responses below, we would like to ask you to follow the line numbers presented in the marked-up version of the manuscript, since lines have changed in the unmarked version after uploading.

Editor comments:

Thank you for your patience with our review of this manuscript during this chaotic time. In conjunction with the previous reviews, I had one additional reviewer look it over and I agree with them that this is a well written paper that, with a few minor edits, is suitable for publication in mSystems. Please make particular note of the comments about the statistical results.

Response: Thank you for handling the manuscript and the anonymous reviewer for the feedback. We have addressed all the reviewer's comments and have reported all the statistical results in supplemental tables (Tables S1-S5).

Reviewer comments:

Reviewer #1 (Comments for the Author):

I commend the authors on a very well planned and executed set of experiments. My comments are uploaded in a word document.

General Comments to the Authors:

In this revised manuscript, Motta & Moran expand on a previous finding that glyphosate can impact honey bee gut microbiota by exploring dose-dependent effects of glyphosate on honey bee gut microbiota. The authors did this by chronically exposing bees to varying concentrations of glyphosate-containing sucrose solution. In addition, they compare the impacts of these treatment groups to a single concentration of a commonly used honey bee antibiotic, tylosin, apparently as a form of positive control. In general, they do find dose-dependent effects of glyphosate exposure on honey bee gut microbiota, although the effects vary greatly with sampling day (5-20), time period (Fall 2018 vs Summer 2019), and the component of the microbial community measured (abundance, diversity, and individual taxonomic units). Still, overall their results do suggest that increasing exposure to glyphosate likely has increasingly direct impacts on honey bee gut microbiota, and potentially indirect effects on survivorship. Although, the latter is speculative, as in a separate experiment they confirm that glyphosate has direct effects on survivorship when applied to bees without established gut microbes, thus they were unable to attribute decreased survivorship to changes in gut microbiotas per se.

Response: The variation reflects real life in outdoor bee hives. It is well established that bees are in different condition in different seasons, and that there is a lot of individual variability among bees within a hive. So, the data do show a lot of individual variation, but the hive experiments have the benefit of realism.

Your comments made us note an interesting aspect of the results, that abundance and alpha diversity are not much affected by the glyphosate treatment, but that taxonomic composition is affected, and in a generally consistent way, as seen in the sharp decreases in *Snodgrassella*. We have added a short paragraph in the discussion to point this out, and to comment on the limitations of standard measures of alpha diversity (lines 383-393).

On your last point above, we see that our explanation was not clear. We have rewritten the Discussion paragraph on the survivorship effects, to emphasize that many sources of mortality in colonies are eliminated in these caged laboratory bees, and that hive mortality rates should be examined to give a full picture (lines 341-356).

I believe this is the second round of revisions based on previous comments, although I did not see an original manuscript file uploaded so am therefore limited in my ability to respond to the suggested revisions by Reviewer #2. I therefore attempt to synthesize the former reviews with my own and present my assessment below. In general, the referenced page numbers in the 'Response to Reviewers Comments' do not align with either 'Manuscript Text File' or 'Marked up Manuscript' which were uploaded for review. My line comments below therefore follow the uploaded versions. I believe for the most part, the cited line revisions do correspond to the highlighted sections in the 'Marked Up Manuscript' document, just not the correct lines. While annoying, I found most responses satisfactory. In general the manuscript was well written, but there were instances where awkward paragraphs lingered or where minor edits were needed. Response: Thanks for these comments. Our revisions in response to the earlier reviewers were extensive, and went beyond simple line edits, as we reworked figures and rearranged the main text. It was too much for 'track changes', so when we uploaded the revised version, we highlighted the sections that were rewritten. We apologize if the line numbers were not correct; we thought that we had checked them. For these responses, please follow the line numbers presented in the marked-up version of the manuscript, since lines have changed in the unmarked version after uploading.

We agree with you that the writing could be polished in numerous places, and we have accordingly edited paragraphs throughout the manuscript. Any substantive edits are mentioned in this Response. We believe that the new version is more readable.

Overall, I agree with the authors that this manuscript will be of much more interest to a broader audience than Reviewer #2 suggested. The authors give many stats from their previous research to back up this point. Moreover, while the target organism of study (honey bees) does have a devoted audience, the broader theory/context behind direct and indirect dose-dependent effects of pesticides on non-target organisms is quite ubiquitous

While the focus of the manuscript is on dose-depend responses, I found it interesting that there was less focus on the implications of their methodological approach, namely that both dose and the timing of exposure to glyphosate, have interactive effects on honey bee gut microbiota and survivorship. Although there was tremendous variation in microbiota response, I think this could use a sentence or two in the discussion

Response: Regarding the effects of stage of exposure and dose, we have expanded on this a little in the discussion (lines 291-295). In fact, responses to glyphosate were surprisingly similar for early and late exposure; for example, *Snodgrassella* showed similar declines with dose, and total bacterial numbers were unaffected for both stages. The impact on community composition, as revealed in the PCoA analyses measured as Bray-Curtis dissimilarity, was evident at lower doses for early exposure (lines 250-253).

Figure 6 C vs D. Why did late exposure bees in the summer of 2019 experience ~ 10 day shorter survivorship overall? It appears from the points of inflection, that overall the experimental replicates experienced the most rapid mortality around 35 days in the early exposure, compared to 25 days in the late exposure. This is a huge difference; was there an explanation I missed?

Response: We are not sure about these differences in survivorship between early and late exposure groups and can only speculate about it. The late exposure bees experienced extra handling, including briefly immobilization with CO₂ after 5 days of gut homogenate treatment, so they could be mixed and randomly assigned to new cup cages to start the chemical treatment.

We believe, but are not sure, that the CO₂ or other handling may have reduced the lifespan of these bees. This is explained in the last paragraph of the subsection “Second chronic exposure of honey bees to glyphosate and tylosin” in the Methods section (lines 462-463). We have now added the following at the end of that paragraph: “*Compared to bees in the early exposure group, bees in the late exposure group experienced an overall increase in mortality probably due to the extra CO₂ exposure and handling*”; and include this information in the results section: “*Overall, bees in the late exposure group experienced elevated mortality rates when compared to bees in the early exposure group, probably due to the extra handling and CO₂ exposure at day 5 when they were reassigned to new cup cages*” (lines 226-229).

Another thing, there were fewer bees in the cups in the later exposure group (30-32 bees), than in the cups in the early exposure group (38-40 bees), which may have also contributed to the differences in survivorship.

Regarding the survivorship data, we have updated Figure 6A-B to include censored bees due to sampling in the model and statistical analyses. These updates did not change conclusions.

One big concern is the lack of reported statistical test results. There are several instances where statistical tests were performed but I was unable to find sufficient results reported in the text of the manuscript or the supplementary material: for example, references to significant p-values for the Kruskal-Wallis tests and Cox Proportional Hazard Models, but no reported Chi-square values, DF, parameter estimates or coefficients. I would highly recommend the authors to verify these exist and are reported appropriately.

Example: On line 535-538: Comparisons of bacterial abundance or alpha diversity were done using Kruskal-Wallis test and Dunn’s multiple comparisons. However, no chi-square results are reported that I can find, either in the manuscript or in the supplementary tables. For example, Figures 3 and 4 show Total and *Snodgrassella* abundances, respectively, and have letters

assigned from the post hoc test, but no chi square. Where are the Kruskal-Wallis tests reported? Shouldn't there be a chi-square value, df, and actual p-values reported? Without them reported I can't in faith accept the post hoc tests.

Response: We forgot to add the statistical reports and apologize for that. They are found now in Supplemental Tables S1, S2, S3, S4 and S5 and have been properly cited in the Results Section and in the relevant Figure legends.

Line Comments:

Line 36. I suggest replacing 'their' with 'honey bee.' Also is 'microbiota' singular or plural? If plural, change to 'microbiota are' throughout. If singular, leave as is.

Response: We accepted these suggestions. 'Microbiota' should be singular, and the plural form should be 'microbiotae'. However, as the field has developed, the latter has rarely been used, and researchers more often use 'microbiotas' for the plural. As suggested, we use it in the singular.

Line 88-92. A 3-sentence introductory paragraph is awkward and detracts from the overall effectiveness of the introduction. Perhaps combine with paragraph 2 about bees?

Response: We agree with the reviewer and have combined paragraphs 1 and 2 and have rewritten so that the central point of the paper is emphasized. We think this will increase the appeal of the article to a broad audience interested in gut microbiomes.

Line 119. Is there evidence of bacterial death in the hindgut due to changes in midgut microbiota? If so references? If not, why?

Response: Our explanation was not clear. Over 95% of the bee gut microbiota is found in the hindgut, which, unlike the midgut, has a stable surface for bacterial attachment and invaginations of the gut wall that facilitate colonization. The midgut microbiota is tiny. Most nutrients acquired from the bee diet, including aromatic amino acids, are absorbed in the midgut, not reaching the region where the microbiota is present. A previous mutagenesis study, showing that *Snodgrassella* requires all amino acid biosynthetic pathways to colonize hosts and implying that amino acids are low in the hindgut, is relevant here. We have rewritten this part, to explain better, and have added this citation (Powell et al. PNAS 2016) (lines 110-113). We refer again to this in the discussion, when we expand the explanation of likely cross-feeding within the community (lines 336-340).

Line 173-174. Interesting, could this suggest increased secondary infection, or just that bacteria are passing through the gut? Should mention why this is important somewhere.

Response: Although we saw this trend in terms of bacterial relative abundance for the early exposure group in the PCA analysis, total absolute values for environmental bacteria were not so clear. We saw an increase in environmental bacterial abundance in 1mM glyphosate-treated bees at day 15, but a decrease at day 20. So, we decided to not highlight this finding as the results are somewhat variable.

Line 220. Is it standard to refer to a Cox Proportional Hazards Model as 'coxph' in the reported stats? I see it is the command for the model in R, but I believe you should list out the actual test

here, not the R command. Also, where are the other model parameter estimates and/or coefficients? Are the reported values only for the significant treatments? There should be p-values and/or coefficients for all reported somewhere. Perhaps a missing Supplementary Table?

Response: We have replaced “coxph” with “Cox Proportional Hazards Model” when appropriate. All the model parameter estimates are now included in Supplemental Table S3. The reported values in this new table compare control to each treatment.

Line 247. Is OTU acceptable without explanation here? I don’t think it was ever defined, but perhaps it was not necessary for the journal?

Response: Since we assigned taxonomy to amplicon sequence variant (ASV), we replaced OTU with ASV and explained the abbreviation the first time it appeared.

Line 253. I agree with Reviewer 2 that traditionally, a PERMANOVA of the centroids is often visualized by ellipses. My bigger concern is that the PERMANOVA is not reported, only a p-value. Where are the other parameters: df, ss, F-stats, etc.?

Response: We added statistical reports for the PERMANOVA analyses in Supplemental Table S4. We tried making the graph with the ellipses, but in our case this addition makes the data very hard to comprehend. So, we prefer to keep it without the ellipses. The Figure 7A with ellipses is shown here, to illustrate:

Line 263. Where are the Chi-squares reported? I only see the letters from the post hoc tests.

Response: We added statistical reports for alpha diversity analysis in Supplemental Table S5.

Line 349. I find these potential bacterial interactions very interesting and think it could be worthwhile to explain in a bit more detail.

Response: We extended this discussion to explain more (lines 336-340).

Line 351-367. This paragraph reads like a list of studies. Maybe revise to link them together for flow? I would suggest focusing on a topic and summary sentence for the paragraph, and selecting the studies that are most relevant to the subject (impacts of glyphosate on survivorship).

Response: We agree and have rewritten this section (lines 341-356). We feel it is important to cite other studies on effects of glyphosate but have telescoped the coverage of this and have related it more specifically to our study, and the question of whether the perturbation of the microbiota affects survivorship of bees. We have moved some information from this paragraph to other parts of the Discussion.

Line 366. There so far has been little to no discussion of synergistic effects from other chemicals. I'd suggest removing this as it doesn't seem relative.

Response: We removed this sentence.

Line 368-371. This 2-sentence paragraph is awkward and seems out of place. Maybe combine with following paragraph about caveats?

Response: We combined this short paragraph with the previous one, as the point about laboratory conditions is central to the discussion of the survivorship results.

Line 390. Did you really test 'acute' vs 'chronic?' Weren't all treatments chronic? Line 435. So were there differences in mortality between the two time periods?

Response: Yes, you are correct, and our wording was not precise. Bees were chronically exposed to chemicals in both experiments. Since we sampled bees at different time points, we considered as "acute" exposure when we sampled bees exposed to glyphosate for only 5 days (first sampling time). For clarification, we rephrased this sentence to: "Such effects occur regardless of the stage of microbiota establishment, and regardless of duration of exposure" (lines 396-397).

Line 545. I think you should explicitly call this a Cox Proportional Hazards Model.

Response: We agree and have replaced.

Line 595. I think this reference to the Kruskal-Wallis test is problematic. You should have one Kruskal-Wallis test reported, if significant, then followed up with the post hoc Dunn's multiple comparisons. Where is the Kruskal-Wallis reported?

Response: We agree with the reviewer and have included all the statistical reports in Supplemental Tables S1-S5.

Additional: We have expanded the detail on the concentrations used or measured in our treatments, expressing them both as mM and as mg/L or mg/kg. This makes our values more

easily compared to values in other studies and helps to prevent confusion. These changes are on lines 156-157 and in the Methods on lines 410-416.

July 8, 2020

Dr. Erick V. S. Motta
University of Texas at Austin
Integrative Biology
2506 Speedway
NMS 4.126
Austin, TX 78712

Re: mSystems00268-20R1 (Impact of glyphosate on the honey bee gut microbiota: effects of intensity, duration and timing of exposure)

Dear Dr. Erick V. S. Motta:

The authors have done an excellent job addressing the reviewer comments.

Your manuscript has been accepted, and I am forwarding it to the ASM Journals Department for publication. For your reference, ASM Journals' address is given below. Before it can be scheduled for publication, your manuscript will be checked by the mSystems senior production editor, Ellie Ghatineh, to make sure that all elements meet the technical requirements for publication. She will contact you if anything needs to be revised before copyediting and production can begin. Otherwise, you will be notified when your proofs are ready to be viewed.

Sincerely,

Sarah Hird
Editor, mSystems

Journals Department
Figure S1: Accept
Figure S4: Accept
Figure S2: Accept
Table S1: Accept
Table S3: Accept
Figure S5: Accept
Table S2: Accept
Table S4: Accept
Figure S3: Accept
Table S5: Accept